# Tracking the CAR-T Revolution: Analysis of Clinical Trials of CAR-T and TCR-T Therapies for the Treatment of Cancer (1997–2020)

**DOI:** 10.3390/healthcare9081062

**Published:** 2021-08-19

**Authors:** Nikola A. Ivica, Colin M. Young

**Affiliations:** 1Koch Institute for Integrative Cancer Research and Department of Biology, Massachusetts Institute of Technology, 500 Main Street, Cambridge, MA 02139, USA; nivica@mit.edu; 2NEWDIGS Initiative, Massachusetts Institute of Technology, 77 Massachusetts Avenue, Cambridge, MA 02139, USA

**Keywords:** CAR-T, TCR-T, cancer immunotherapy, biologics, clinical trials

## Abstract

Chimeric antigen receptor and T-cell receptor (CAR-T/TCR-T) cellular immunotherapies have shown remarkable success in the treatment of some refractory B-cell malignancies, with potential to provide durable clinical response for other types of cancer. In this paper, we look at all available FDA CAR-T/TCR-T clinical trials for the treatment of cancer, and analyze them with respect to different disease tissues, targeted antigens, products, and originator locations. We found that 627 of 1007 registered are currently active and of those 273 (44%) originated in China and 280 (45%) in the US. Our analysis suggests that the rapid increase in the number of clinical trials is driven by the development of different CAR-T products that use a similar therapeutic approach. We coin the term bioparallels to describe such products. Our results suggest that one feature of the CAR-T/TCR-T industry may be a robust response to success and failure of competitor products.

## 1. Introduction

Chimeric antigen receptor and T-cell receptor (CAR-T/TCR-T) immunotherapies are targeted cellular therapies that use the cytotoxic potential of T cells to eradicate cancer cells in an antigen-specific manner [1,2,3]. The therapeutic approach involves genetic modification of isolated T cells from a patient in order to express the desired CAR or TCR gene on cells’ surface (Figure 1A). Genetically modified T cells are subsequently infused back into the patient, where they eventually come in direct contact with the cancer antigen, resulting in the killing of the cancer cell [4]. CAR-T cells are able to specifically recognize antigens on the surface of cancer cells via an extracellular single-chain fragment variable (scFv) domain, which upon engagement with the cancer antigen results in immobilization and clustering of CARs and formation of non-classical immune synapse, leading to the activation of the T cell [5,6,7]. TCR-T cells use TCR to recognize cancer-associated antigen in the context of major histocompatibility complex-I (MHC-I). MHC-I is found on the surface of all nucleated cells where it presents peptides derived from proteolytic processing of intracellular proteins. TCRs can specifically recognize cancer-associated peptides in the context of MHC-I and activate the T cell upon engaging with the complex. Activated T-cell have pleiotropic functions, which include rapid proliferation, killing of cancer cells, and release of cytokines. Several mechanisms have activated T cells use to kill cancer cells. Secretion of cytotoxic granules containing perforin and granzyme directly kills cancer cells via apoptosis [8]. In addition, activated T cells express death receptors on their surface that bind to death receptor ligands on cancer cells, resulting in their death [9,10,11]. T cell activation also results in secretion of multiple cytokines, such as IL-2, IL-6, IFN-γ, and GM-CSF, which recruit and activate other immune cell types, most notably macrophages and natural killer cells.

In 2017, the Food and Drug Administration (FDA, Silver Spring, MD, USA) approved two CAR-T therapies, tisagenlecleucel (Kymriah^®^) and axicabtagene ciloleucel (Yescarta^®^), for the treatment of acute lymphocytic leukemia and diffuse large B-cell lymphoma, respectively. Recent long-term follow-up studies for lymphoma [12,13,14] showed overall response rates (ORR) >50% with complete response (CR) of 37% and 40% at 12 and 27 months, respectively. These numbers represent a major improvement in treatment efficacy and response durability, especially for those patients who are refractory to chemotherapy. More recently, in 2020, the FDA approved a CAR-T therapy for mantle cell lymphoma, brexucabtagene autoleucel (Tecartus^®^). (Subsequent to our sample cutoff date, two further CAR-T therapies received FDA approval-lisocabtagene maraleucel (Breyanzi^®^) for adult large B-cell lymphoma and idecabtagene vicleucel (Abecma^®^) for multiple myeloma).

In light of the recent success (and future potential) of CAR-T/TCR-T therapies for the treatment of cancer, we aim to provide a comprehensive overview of the growth of the clinical trials space for these products, and to highlight some of the factors driving this growth. One such factor is the nature of CAR-T/TCR-T immunotherapies, which makes it possible to develop many different products that use analogous therapeutic approach to achieve comparable clinical efficacy [15,16]. Such CAR-T/TCR-T products that are currently in the clinical pipeline cannot be characterized as biosimilar products in the narrow sense of the term, so we propose that CAR-T/TCR-T products that use a parallel therapeutic approach be termed bioparallels. For example, Kymriah and Yescarta are both CD19-targeted CAR-Ts and both use scFv FMC63 but different promoters (EF1a vs. MSCV), co-stimulatory (4-1BBzeta vs. CD28zeta) and hinge and transmembrane domains (CD8α vs. CD28). In Figure 1B,C, we depict several ways of developing bioparallel products. We identified a total of 1007 registered clinical trials related to CAR-T/TCR-T product development. Segregation of trials by cancer tissue, and antigens targeted, reveal a rapid rise in the development of bioparallel products in the past 7 years that primarily concentrate on replicating therapeutic approaches with lower risks of failure in the clinic.

## 2. Data Sources

Information related to CAR-T/TCR-T clinical trials was collected from the NIH’s ClinicalTrials.gov (CTgov, Bethesda, MD, USA). Only trials registered in CTgov are included in our sample (henceforth “CTgov” trials). The dataset is available as a Appendix A). In total, we identified 1007 NCT trials that involve CAR-T/TCR-T therapies or close analogs. Of these, 774 were CAR-Ts and 153 were TCRs. The remaining 80 trials comprised bispecific CAR CIK cells (cytokine-induced killer cells), CAR NK cells (natural killer cells), and T cells engineered to have properties normally found in NK cells. Because the start date of a trial, as reported on CTgov, typically represents the starting point for patient enrollment, we used this date as the clinical trial initiation date in our analysis. Among the trials, we found 50 that were flagged as “Not yet recruiting”—these were eliminated from our analysis, leaving a sample of 957 initiated trials. The first clinical trial in our database started in February 1997 (NCT00019136), with our sample including all identified CTgov trials starting between that date and 31 December 2020 (details of these trials may be found in Appendix A).

## 3. Cellular Immunotherapies for Cancer: Overview of Clinical Trials

### 3.1. Clinical Trials for CAR-T/TCR-T and Related Therapies

Of the 957 initiated clinical trials in our sample, 233 are no longer active (24.3%) and 147 have an “Unknown” status (15.3%). Of the 627 remaining active clinical trials, 390 trials (40.7%) are in Phase 1, 221 trials (23.1%) are in Phase 1/2 or Phase 2, 10 are in Phase 2/3 or Phase 3, and the rest of the trials are either long term follow ups (7) or trials with no self-reported phase (4) (Table 1).

Annual and cumulative clinical trials initiations for cellular immunotherapies exhibited steadily accelerating growth through 2020 (Figure 2A). For comparison, and to provide some context, Figure 2B shows the number of initiated clinical trials for anti-PD-1 (Keytruda, Opdivo) and anti-PD-L1 (Tecentriq, Imfinzi, Bavencio) antibody therapies for the same period. A total of 3352 trials of these therapies were initiated through 2020. In combination, the cellular immunotherapy and antibody trials initiated constituted less than 10% of the 46,425 CTgov interventional cancer trials initiated between 2008 and 2020. While representing a small proportion of total trials, cellular immunotherapies address a much larger portion of total cancer incidences. Surveillance, Epidemiology, and End Results (SEER, Bethesda, MD, USA) program data indicates that 5-year mortality for cancer as a whole is around 32% (for specific cancers it is much higher). These patients, refractory and/or relapsed with respect to other therapies, form the potentially treatment eligible populations for cellular immunotherapies (details of these populations for the specific cancers addressed by the trials in our sample may be found in Appendix A).

We note that slowdowns in CAR-T/TCR-T trials initiations apparent in Figure 2A appear to have occurred at about the time of some widely reported failures including the failure of a TCR-T trial in colorectal cancer in 2009 [17] and the failures of two TCR-Ts targeting MAGE-A3 in solid tumors in 2011 [18,19]. There was no such slowdown coincident with the widely publicized termination of Juno’s JCAR015 in 2016. The remainder of this section provides a number of breakdowns of the overall picture of trials initiation. Our results are generally consistent with previous analyses of TCR-Ts [20] and CAR-Ts [21], and of the pipeline of cancer cell therapies [22], which were based on earlier data.

While the number of initiated trials has increased for all therapies, the number of CAR-T trials has increased significantly faster, especially since 2013–CAR-T trials initiations grew at a 38.7% compound annual growth rate vs. 25.6% for the other therapies in our sample from 2013 to the end of 2020 (henceforth E2020) (Figure 3A). That growth was divided equally between CAR-Ts targeting CD19–the antigen targeted by Kymriah- and CAR-Ts targeting all other antigens (Figure 3A). Within those other antigens, we can also observe the increase in trials targeting BCMA after 2016 (from a total of seven initiated trials at that point to 42 in 2018 and 85 E2020). This followed encouraging results in trials treating multiple myeloma (NCT02658929, NCT03090659). Overall, the result has been that, while the initiation of trials targeting solid tumors had kept pace with those for hematologic cancers through 2015, 2016-E2020 saw initiation of 479 trials targeting hematologic cancers (64%), compared to 265 targeting solid tumors. (Figure 3C). One factor in the overall increase in hematologic trials’ initiations was the increase in the number of trials initiated by China-based originators (Figure 3D).

Our sample includes trials originated by sponsors in 21 countries (Table 2). Originators based in the United States (408 trials) and China (433) dominate, with the United Kingdom (43) a distant third. The other 18 countries originated 73 trials between them. The first CTgov trial from a China-based originator was initiated in 2012. In the following two years a further 13 trials were initiated. Since that date a further 419 CTgov trials have been initiated by China-based originators comprising almost 52% of all trials initiated over that period. Breaking it down further, in 2016-E2020, China-based sponsors initiated 267 trials targeting hematologic malignancies—55.7% of such trials initiated in that period. Finally, we also note that China-based originators have focused very strongly on CAR-T development (more than 86.4% of initiated trials) in contrast to US-based originators for whom TCR-Ts and other related therapies made up more than 26% of trials initiated (Figure 4A,B).

### 3.2. Tissues and Antigens Targeted

As previously noted, particularly in 2016-E2020, trials initiated for hematologic cancers far exceeded those for solid tumors with most of the hematologic trials (90%) being CAR-Ts targeting either B-cell leukemias and lymphomas (404 out of 529 trials with 368 targeting CD19) or multiple myeloma (99 trials of which 85 target BCMA) (Figure 5A–C). Other hematologic cancer trials target myeloid leukemias or myelodysplasias (58 trials), as well as T-cell lymphomas and Hodgkin’s lymphoma (26 trials). While predominantly CAR-Ts, these also include TCR-Ts and other related therapies.

Trials addressing solid tumors comprise CAR-Ts (212), TCR-Ts (125) and other related therapies (33) (Figure 6A). They target over 60 antigens or oncoproteins expressed by tumors affecting 20 solid organs and tissues. The most frequently targeted antigens are NY-ESO-1 (whose 54 trials make up 43% of all solid tumor TCR-Ts), mesothelin (33), EGFRvIII (29), GD2 (33), HER2 (24), and MUC1 (24) (Figure 6B). The most frequently targeted tissue types have been cancers of the brain and nerve tissue (52 trials), liver (42), female reproductive system (37), and lung (31) (Figure 6C). The number of multi-organ CAR-T/TCR-T clinical trials that recruit patients with different solid tumors has increased rapidly since 2013 (currently 82 trials), likely because it is a more efficient strategy to perform trials and recruit patients.

## 4. Discussion

### 4.1. Rapid Rise in the Number of CAR-T/TCR-T Bioparallel Products

We define bioparallel products as different biological products that use a parallel therapeutic approach to achieve similar clinical efficacy. In the case of CAR-T/TCR-T therapies, bioparallel products can be designed in several ways. Figure 1B,C depict how several different CAR-T/TCR-T products can target the same antigen by recognizing a different epitope. For example, the two FDA approved CAR-T therapies, tisagenlecleucel (Kymriah) and axicabtagene ciloleucel (Yescarta), both target the same antigen, CD19. In the case of TCR-T therapies, JTCR016 (Juno Therapeutics, Seattle, WA, USA) and CMD-602 (Cell Medica Ltd., London, UK) are examples of two different TCR-T products undergoing clinical trials that recognize the same antigen, Wilms tumor 1 (WT-1), which is expressed in multiple types of cancer. Our clinical trials analysis suggests that the development of bioparallels may be driving the overall numbers of CAR-T/TCR-T trials as many companies are in the process of entering the market by developing their own variants of a therapeutic approach that has a lower risk of failure or has already proven successful. CAR-T products that target CD19 are the most obvious example-we have identified over 100 bioparallel CD19 targeting products that are either in trials or are already approved (with around 20 more in multi-CAR-T trials). Another example is a CAR-T therapeutic approach that targets BCMA for the treatment of multiple myeloma. BCMA is a cancer antigen that is therapeutically similar to CD19 in several ways. It is highly expressed on all plasma cells and loss of BCMA does not have significant adverse influence on the overall homeostasis of B-cells, but it is critical for long-term survival of plasma cells [23]. This means that eradicating BCMA expressing cells using CAR-T/TCR-T approach is a sound therapeutic option. We have identified 85 clinical trials that target BCMA, and CAR-T clinical trials targeting this antigen are the fastest growing after CD19. Current clinical trial pipelines consist of over 40 different BCMA CAR-T programs.

In addition to targeting different epitopes, the highly modular nature of CAR-T therapeutics allows for even greater product diversification, as was evident in the development of several different generations of CAR-T designs [4]. Some approaches target multiple epitopes with the same CAR-T product. For example, several clinical trials use bispecific CAR-T cells that simultaneously target CD19 and CD20 or CD22 (NCT03271515, NCT03241940). Expression of recombinant receptors, such as EGFRt, for antibody-mediated depletion of CAR-T cells from the patient, or the use of T cells with deleted PD-1 to avoid immunosuppression, are more recent CAR-T products (NCT03085173, NCT03298828). In addition, a 2019 phase 2 study used a combination of anti-CD19 and anti-BCMA CAR-T cells in relapsed or refractory multiple myeloma patients [24]. In total, 22 patients received 10^6^ cells/kg of each human anti-CD19 and murine anti-BCMA CAR-T cells. Overall response was 95%, while 43% of patients had a stringent complete response, suggesting that CAR-T combination therapies may be another source of product diversification. These and other variants of CAR-T/TCR-T products represent a platform for the development of an ever-growing number of bioparallels [25]. Indeed, this behavior resembles to a large extent the “gold rush” of biosimilar monoclonal antibodies, which can be diversified to a similar extent [26,27].

### 4.2. Effects of Clinical Success or Failure in the CAR-T/TCR-T Market

The rapid rise in the number of developing CD19 CAR-T products after reports of clinical success from tisagenlecleucel trials in 2012 is an indication of how the market responds to a successful therapeutic approach in the CAR-T/TCR-T space. Similarly, the recent increase in the number of BCMA CAR-T products for the treatment of multiple myeloma represents another example of how a successful clinical trial may influence development decisions at other companies. In 2016, two clinical trials of BCMA targeting CAR-Ts reported therapeutic tolerability and efficacy, with more than half of the patients achieving complete remission (NCT02658929, NCT03090659). In the subsequent four years, 78 BCMA trials were initiated.

Developers are not only connected because of the intrinsic biological similarity of their products, but there seems to exist a complex network of CAR-T/TCR-T companies that are interacting via technology licensing, academic collaborations, and research support. Indeed, the cost of research and development of a new drug presents a risk for a pharmaceutical company, so leveraging the information from other similar trials can alleviate some of the costs [28]. Therefore, the reports of clinical success or failure of one product in development would have an effect on a more global scale. While this is not a novel behavior in drug development industry, different CAR-T/TCR-T therapies are so biologically similar that trends appear to be more robust.

Finally, the observation of relatively fewer different CAR-T/TCR-T products and clinical trials for solid tumors may be explained by the lack of clinical efficacy for current products and the previously noted failures of several earlier TCR-T trials. The greatest number of clinical trials has been initiated for cancers affecting brain and nerve tissue, for which CAR-T/TCR-T therapies have not yet demonstrated clinical success. Clinical trials have thus far focused on CAR-T products targeting the GD2, EGFRvIII, and HER2 antigens and TCR-Ts targeting NY-ESO-1, a cancer/testis antigen. GD2 is a complex glycosphingolipid, highly expressed on the surface of several solid tumors, especially neuroblastoma and melanoma, while EGFRvIII and HER2 are surface protein receptors highly expressed on the surface of many solid tumors. These antigens were known from previous studies to be clinically suitable targets for monoclonal antibody therapy [29,30]. In the context of CAR-T therapy, all three antigens have shown clinical tolerability; however, none of the approaches has resulted in significant eradication of tumor cells [31,32,33]. The proposed reasons for poor performance of CAR-T/TCR-T therapies in solid tumors range from the highly immunosuppressive environment, to the lack of therapeutically effective cancer antigens [34,35,36]. Were these roadblocks to be effectively overcome, it would not be surprising to see a growth in the number of initiated trials similar to that experienced in trials targeting BCMA or CD19.

## 5. Conclusions

CAR-T and TCR-T therapies are revolutionary biotechnology products that have already shown long-term durable remission for the treatment of some types of refractory B-cell leukemias and lymphomas, with potential to provide durable remission for other types of cancers. Our analysis of all identified CTgov trials for CAR-T/TCR-T therapeutics suggests that there is a rapid rise in the number of bioparallel products, the term we coined to describe different products that use a parallel therapeutic approach. It will be interesting to see if this continues in the future as more CAR-T/TCR-T products get FDA approval, particularly if one is a CAR-T/TCR-T therapy for the treatment of a solid tumor.

## Figures and Tables

**Figure 1 healthcare-09-01062-f001:**
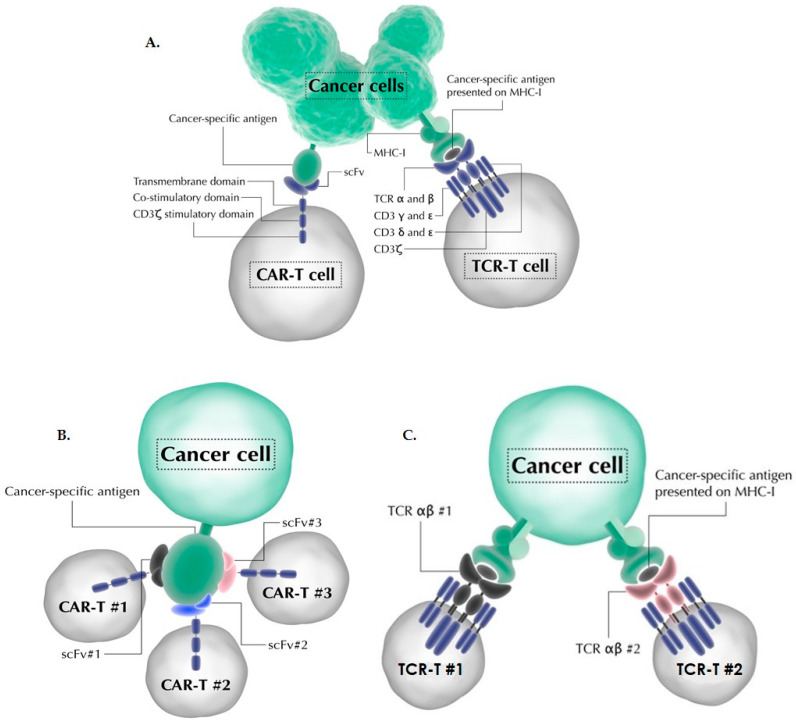
Schematics of common CAR-T and TCR-T cellular immunotherapy products. (**A**) Components of common chimeric antigen receptor (CAR) and T-cell receptor (TCR) used in clinical trials. CAR-T cells express a chimeric receptor on their surface composed of a single-chain fragment variable (scFv) domain, a transmembrane domain, and co-stimulatory and stimulatory domains. The scFv domain binds to cancer-associated antigens on the surface of cancer cells. The transmembrane domain positions the receptor on the cell surface, while co-stimulatory (usually CD28 or 4-1BB fragments) and stimulatory (CD3ζ fragment) domains activate the T cell, resulting in cytotoxicity and proliferation. Cancer-associated antigens targeted by CAR-T cells are typically differentiation antigens or other antigens that are over-expressed by tumor cells. TCR-T cells are characterized by the presence of a TCR composed of α and β chains. TCR binds specifically to a small peptide presented by major histocompatibility complex-I (MHC-I), and upon binding TCR activates the T cell via endogenous CD3 signaling. Most commonly, TCR-Ts target cancer/testis antigens (CTAs)-antigens that are expressed only by cancer cells, germ cells, and placental tissue. These antigens are expressed inside the cell and require MHC-I to present them on the cell surface. A TCR-T must not only target the CTA, but also be compatible with the presenting MHC-I. (**B**) Different CAR-T cell products can target the same antigen. For example, two different scFvs can bind to different parts of the cancer-associated antigen. We term these products bioparallels. (**C**) Similarly, different TCR-T products can target the same antigen in the context of MHC-I giving rise to potentially many different TCR-T bioparallel products.

**Figure 2 healthcare-09-01062-f002:**
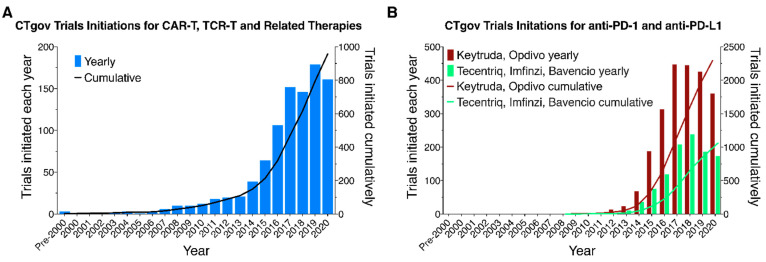
Yearly and cumulative numbers of CAR-T/TCR-T clinical trials registered on clinicaltrials.gov and initiated on or before 31 December 2020, under FDA guidelines. (**A**) CAR-T/TCR-T and related cellular immunotherapies. (**B**) PD-1 (Keytruda, Opdivo) and PD-L1 (Tecentriq, Imfinzi, Bavencio) antibody therapies.

**Figure 3 healthcare-09-01062-f003:**
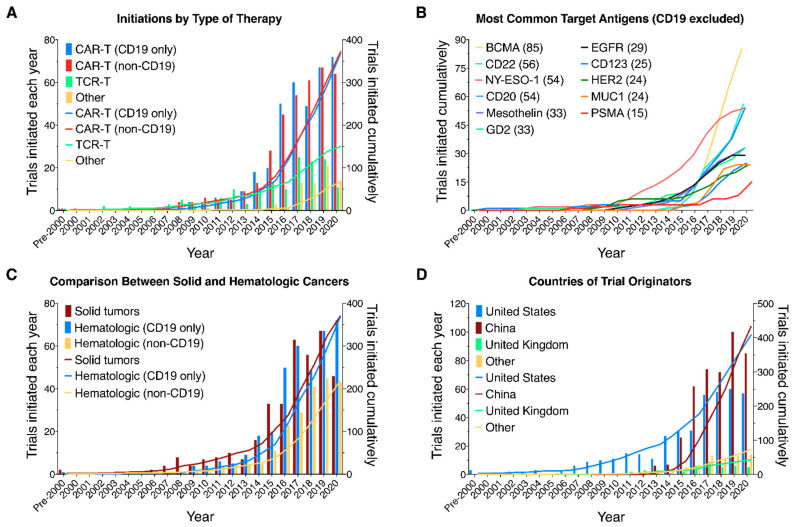
Comparison of different types of therapy, antigens, and tissue targeted, and countries of clinical trial originators. Yearly (bars) and cumulative (lines) numbers of CAR-T/TCR-T trials registered on clinicaltrials.gov and initiated on or before 31 December 2020. (**A**) Trials segmented by type of therapy. CAR-Ts are divided into those targeting CD19 (either alone or in combination with other antigens) and all other trials. (**B**) Number of clinical trials for the 11 most frequently targeted antigens other than CD19. (**C**) Comparison of clinical trials targeting solid tumors and hematologic cancers. Hematologic cancer trials are divided into those targeting CD19 (either alone or in combination with other antigens) and all other trials. (**D**) Number of clinical trials by originators from China, United States, United Kingdom, and other countries.

**Figure 4 healthcare-09-01062-f004:**
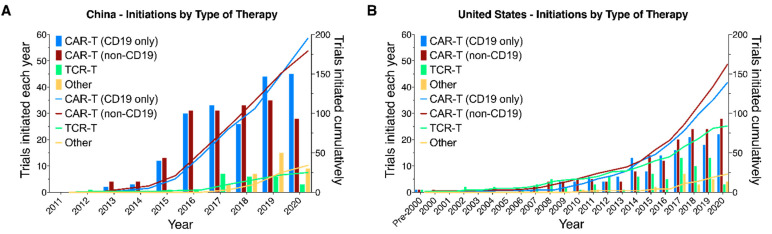
Number of CAR-T clinical trials by originators from China surpassed the number of clinical trials by originators from the United States. Yearly (bars) and cumulative (lines) numbers of CAR-T/TCR-T trials registered on clinicaltrials.gov and initiated on or before 31 December 2020. CAR-Ts are divided into those targeting CD19 (either alone or in combination with other antigens) and all other trials. (**A**) China-based and (**B**) United States-based clinical trial originators.

**Figure 5 healthcare-09-01062-f005:**
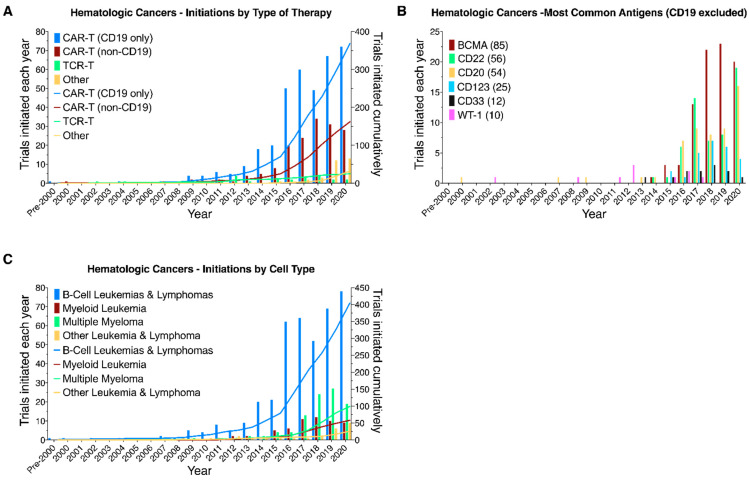
CAR-T/TCR-T clinical trials targeting hematologic cancers. Yearly (bars) and cumulative (lines) numbers of CAR-T/TCR-T trials registered on clinicaltrials.gov and initiated on or before 31 December 2020, targeting hematologic cancers. Trials were segmented by (**A**) type of therapy, (**B**) commonly targeted antigens, and (**C**) targeted cell type.

**Figure 6 healthcare-09-01062-f006:**
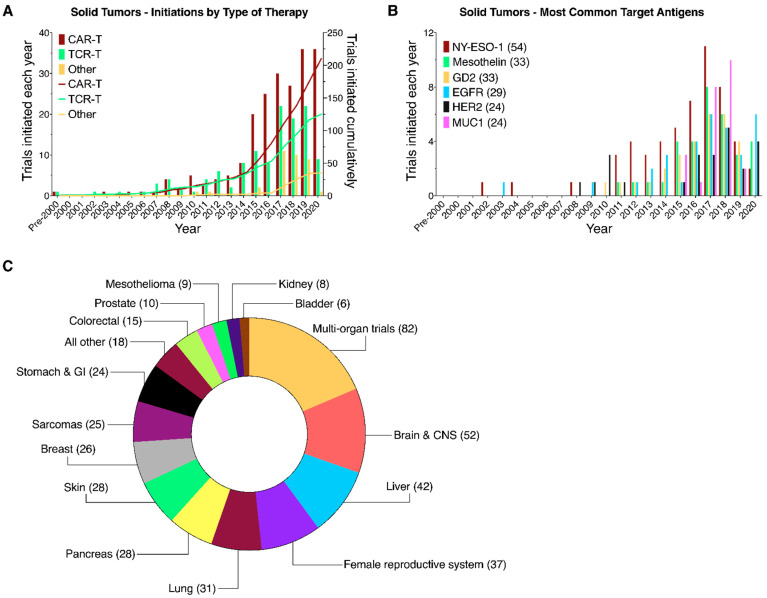
CAR-T/TCR-T clinical trials targeting solid tumors. Yearly (bars) and cumulative (lines) numbers of CAR-T/TCR-T trials registered on clinicaltrials.gov and initiated on or before 31 December 2020 targeting hematologic cancers. Trials were segmented by (**A**) type of therapy, (**B**) commonly targeted antigens, and (**C**) the tissue of origin.

**Table 1 healthcare-09-01062-t001:** Phase and status of all CAR-T/TCR-T trials registered on clinicaltrials.gov with start dates on or before.

NCT Trials Classifications	Early Phase 1	Phase 1	Phase 1 Phase 2	Phase 2	Phase 2 Phase 3	Phase 3	Not Applicable	Total
Not yet recruiting	18	22	10	0	0	0	0	50
Recruiting	48	262	142	38	3	6	2	501
	7.7%	42.1%	22.8%	6.1%	0.5%	1.0%		
Active, not recruiting	3	74	26	16	0	1	1	121
	0.5%	11.9%	4.2%	2.6%	0.0%	0.2%		
Enrolling by invitation	1	2	0	0	0	0	2	5
	0.2%	0.3%	0.0%	0.0%	0.0%	0.0%		
Completed	4	49	14	9	1	1	3	81
Withdrawn	0	15	11	11	0	1	0	38
Terminated	6	21	10	12	0	0	1	50
Suspended	0	7	5	2	0	0	0	14
Unknown	15	59	63	6	3	0	1	147
Total	95	511	281	94	7	9	10	1007

The cutoff date is 31 December 2020. For known active trials, percentages of the total of applicable active trials (622) are given.

**Table 2 healthcare-09-01062-t002:** For each originating country, numbers of CAR-T/TCR-T trials under.

Country of Origin	Total Trials	Initiated	Currently Active	Currently Inactive
China	470	433	273	197
US	418	408	280	138
United Kingdom	43	43	26	17
Germany	15	15	11	4
Belgium	10	10	7	3
France	8	8	4	4
Japan	7	7	4	3
Singapore	9	7	3	6
Australia	3	3	2	1
Italy	4	4	4	0
Netherlands	3	3	2	1
Spain	3	3	3	0
Israel	2	2	2	0
Russia	2	2	2	0
Sweden	2	2	1	1
Switzerland	2	2	0	2
Canada	2	1	1	1
Czech Republic	1	1	0	1
Malaysia	1	1	1	0
Norway	1	1	0	1
Turkey	1	1	1	0
Total	1007	957	627	380

FDA guidelines listed, initiated and currently active as of 31 December 2020.

## Data Availability

Data are contained within the article or supplementary material. The data presented in this study are available online at www.mdpi.com/xxx/S1. Raw data were extracted from www.clinicaltrials.gov (most recently accessed on 6 July 2021).

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
