# Peer review of "Tracking the CAR-T Revolution: Analysis of Clinical Trials of CAR-T and TCR-T Therapies for the Treatment of Cancer (1997–2020)"

_healthcare, 2021, doi:10.3390/healthcare9081062_

Round 1

Reviewer 1 Report

In this manuscript, Ivica and Young proposed a complete analysis on trials related to CAR-T and TCR therapies in the last 20 years. The authors provided extensive data regarding the available information on number of trials, antigens used, countries of origin, etc. that overall, represented an informative and interesting study on a hot topic within cancer therapy.  Remarkably, plots were very illustrative and easy to interpret.

Nevertheless, the article would substantially benefit from the following comments:

  1. I suggest to change either the format to make it fit more into a Review manuscript (by changing the title of sections 2 and 3, possibly subdividing section 3) or considering the article as an analytical/research type one.
  2. The introduction should include a more detailed explanation on the mechanism of action of CAR-T and TCR-T products.
  3. Figure 1 misses the explanation on A) panel.
  4. Please change all over the text, but in particular in Figure 1 design, TCR by TCR-T as this appears confounding for people outside the field. TCR is a functional receptor and TCR-T cells are T cells modified with engineered TCRs, this is very important to clarify. Again, in the figure 1 legend there is a clear confusion in that sense.
  5. Related with previous point, In Figure 1 legend, please replace “CAR-T cell is composed of an cFV domain” by “CAR-T cell is characterized by the presence of an cFV domain” or something similar because CAR-T cells are not receptors nor domains. Similar comment applies to” TCR is composed of” that should be replaced by “TCR-T cells are characterized by”.
  6. A plot comparing the number of clinical trials based on these technologies in confront to all the rest of therapeutic cancer trials or similar ( based on total numbers or year increments, etc.) would be informative on the relevance of those trials for cancer therapy.
  7. Please change colors of Figure 6C as they are many blue sectors ( you can use lilac, violet, pink instead) and increase letter size.
  8. Please comment and cite the following article that provides information both about the biological bases of CAR-T/TCR-T therapies and about related clinical trials: Zhao and Cao, 2019 Frontiers in Immunology, doi:10.3389/fimmu.2019.02250

Reviewer 2 Report

The objective of this study was to identify the current available FDA CAR-T/TCR clinical trials and analyze these trials based on different aspects including disease site, targeted antigens, products, and originator locations. The authors reported that they collected information related to CAR-T/TCR clinical trials from the NIH’s ClinicalTrials.gov and derived a final dataset comprised of 1007 trials and conducted a review and analysis of these trials.

Thank you for giving me the opportunity to review and critique this work. It was interesting and the authors did well in collecting the information about the clinical trials but unfortunately, I find that the novelty and the significance of the work are low. The analysis of these trials was only a description of the main aspects of the trials. There were a lot of figures presented but I do not see a benefit from the inclusion. Overall, I do not think this article is suited for healthcare as a review paper.

Author Response

We thank Reviewer 2 for taking the time to read the manuscript and provide comments. The paper provides an up to date historical perspective on the development of CAR-Ts and TCR-Ts as a treatment modality. As a review, it assembles and presents information, which in some cases is already available from other sources, in a comprehensive and easily accessible format.

Reviewer 3 Report

in this review the authors try to analyze the recent development in the field of CAR-T/TCR therapies for the treatment of some resistant B-cell malignancies. The paper could be interesting for the researcher in this specific research area. The authors could increase the quality of the work maybe by better describing the different trials in the results and not just by listing how many and which ones have been active in recent years. The authors in the discussion could better emphasize the differences between the trials. 

Author Response

We thank Reviewer 3 for taking the time to read the manuscript and for providing us with useful comments. We feel that we have covered a wide range of trials targeting many types of malignancy. We have expanded details of methods of action of CAR-Ts and TCR-Ts and somewhat rewritten the material on the biological bases of the therapies. The manuscript now contains a more in-depth analysis of several clinical trials and highlights differences between them.